# Evaluation of the Biocompatibility and Endothelial Differentiation Capacity of Mesenchymal Stem Cells by Polyethylene Glycol Nanogold Composites

**DOI:** 10.3390/polym13234265

**Published:** 2021-12-06

**Authors:** Huey-Shan Hung, Yi-Chin Yang, Wei-Chien Kao, Chun-An Yeh, Kai-Bo Chang, Cheng-Ming Tang, Hsien-Hsu Hsieh, Hsu-Tung Lee

**Affiliations:** 1Graduate Institute of Biomedical Science, China Medical University, Taichung 40402, Taiwan; hunghs@mail.cmu.edu.tw (H.-S.H.); doraemon_blue1201@hotmail.com (W.-C.K.); hs0603@gmail.com (C.-A.Y.); kbwork2021@gmail.com (K.-B.C.); 2Translational Medicine Research, China Medical University Hospital, Taichung 40402, Taiwan; 3Department of Neurosurgery, Neurological Institute, Taichung Veterans General Hospital, Taichung 407204, Taiwan; jean1007@gmail.com; 4College of Oral Medicine, Chung Shan Medical University, Taichung 40201, Taiwan; ranger@csmu.edu.tw; 5Blood Bank, Taichung Veterans General Hospital, Taichung 407204, Taiwan; hhhsu@vghtc.gov.tw; 6Cancer Prevention and Control Center, Taichung Veterans General Hospital, Taichung 407204, Taiwan; 7College of Medicine, National Chung Hsing University, Taichung 402, Taiwan; 8Graduate Institute of Medical Sciences, National Defense Medical Center, Taipei 11490, Taiwan

**Keywords:** polyethylene glycol, gold nanoparticles, mesenchymal stem cells, endothelialization, vascular regeneration

## Abstract

Cardiovascular Diseases (CVDs) such as atherosclerosis, where inflammation occurs in the blood vessel wall, are one of the major causes of death worldwide. Mesenchymal Stem Cells (MSCs)-based treatment coupled with nanoparticles is considered to be a potential and promising therapeutic strategy for vascular regeneration. Thus, angiogenesis enhanced by nanoparticles is of critical concern. In this study, Polyethylene Glycol (PEG) incorporated with 43.5 ppm of gold (Au) nanoparticles was prepared for the evaluation of biological effects through in vitro and in vivo assessments. The physicochemical properties of PEG and PEG–Au nanocomposites were first characterized by UV-Vis spectrophotometry (UV-Vis), Fourier-transform infrared spectroscopy (FTIR), and Atomic Force Microscopy (AFMs). Furthermore, the reactive oxygen species scavenger ability as well as the hydrophilic property of the nanocomposites were also investigated. Afterwards, the biocompatibility and biological functions of the PEG–Au nanocomposites were evaluated through in vitro assays. The thin coating of PEG containing 43.5 ppm of Au nanoparticles induced the least platelet and monocyte activation. Additionally, the cell behavior of MSCs on PEG–Au 43.5 ppm coating demonstrated better cell proliferation, low ROS generation, and enhancement of cell migration, as well as protein expression of the endothelialization marker CD31, which is associated with angiogenesis capacity. Furthermore, anti-inflammatory and endothelial differentiation ability were both evaluated through in vivo assessments. The evidence demonstrated that PEG–Au 43.5 ppm implantation inhibited capsule formation and facilitated the expression of CD31 in rat models. TUNEL assay also indicated that PEG–Au nanocomposites would not induce significant cell apoptosis. The above results elucidate that the surface modification of PEG–Au nanomaterials may enable them to serve as efficient tools for vascular regeneration grafts.

## 1. Introduction

Blood vessels such as arteries, veins, and capillaries are necessary for blood flow in the human body [1]. The cardiovascular system is responsible for oxygen, carbon dioxide, and nutrient transportation within the human body [2]. Therefore, the body’s cardiovascular system is essential for life. According to statistical analysis from the World Health Organization, Cardiovascular Diseases (CVDs) caused 32% of deaths worldwide in 2019 [3]. CVDs include stroke, myocardial infarction, coronary thrombosis, and atherosclerosis, along with other diseases [4]. In clinical approaches, vascular replacement surgery is the major form of treatment for patients with CVD. Thus, vascular grafts have become the primary materials for use in patients with cardiac artery failure. Synthetic polymers such as Polytetrafluoroethylene (PTFE) are reported to be safe vascular graft options for severe atherosclerosis [5]. However, a previous study has indicated that these vascular grafts tend to induce an atherosclerosis-like phenomenon in spite of showing success in graft replacement [6]. The reason for this may be attributed to lipid retention, which is influenced by the strong affinity of the hydrophobic PTFE polymer [7]. Moreover, neointimal hyperplasia, thrombosis, and interstitial calcification at the implantation site commonly leads to grafting failure [8], thus owing to poor biocompatibility and inefficiency of endothelialization [9]. Therefore, various surface modification approaches have been of high concern and thus well investigated for purposes of improving the hemocompatibility and the endothelial differentiation capacity of the blood-contacting surface [10,11]. 

Improving endothelialization and preventing thrombosis in artificial blood vessel grafts are major tasks surrounding vascular regeneration engineering [12]. Various types of synthetic polymers such as Polyurethane (PU) and Poly(Lactic-co-glycolic acid) (PLGA) have been well investigated for use in tissue repair. PU has been widely used in medical applications, owing to better blood compatibility and mechanical properties for mimicking natural vascular tissue [13]. PLGA has been verified to be a biocompatible and biodegradable polymer for medical utilization [14]. Furthermore, PEG has also been widely applied for the surface modification of artificial grafts due to its decreasing cytotoxicity and enhanced biocompatibility [15]. The available literature has indicated that the degradable PLGA/collagen nanofibers, modified by Mesoporous Silica Nanoparticle (MSN) grafting with PEG and heparin, can enhance blood compatibility and cell proliferation to maintain long-term vascular patency [16]. Another study demonstrated that a Polycaprolactone–Polyethylene Glycol Methyl Ether (PCL-PEGME) mat could induce the endothelial gene CD31 for expression [17]. Above all, PEG fabricated scaffolds demonstrated themselves to be a potential material for vascular repair engineering. 

Nanoparticles such as gold (Au) have been extensively investigated and used for biomedical applications such as drug delivery [18] and regenerative medicine [19]. Au is a metal with both stability and superior biocompatibility [20], and it can be immobilized with biomolecules such as proteins and enzymes [21]. Moreover, Au nanoparticles can be easily controlled at different sizes with structures manufactured by Surface Plasmon Resonance affecting cytotoxicity and biological functions [22]. According to our previous research, Au successfully incorporated itself with polymers such as Polyurethane (PU) [23], Fibronectin (FN) [24], and type I Collagen (Col) [25]. For instance, the surface modification of PU combined with Au (PU–Au) demonstrated that it could significantly activate FAK and PI3K/Akt signaling pathways in Endothelial Cells (ECs), which then affected the ECs’ proliferation and migration ability [23]. Furthermore, the coating surface of Fibronectin fabricated with Au (FN–Au) resulted in better biocompatibility than inhibiting monocyte activation and platelet activation. Additionally, FN–Au nanocomposites also influenced the cell behavior of human umbilical cord-derived MSCs. FN containing 43.5 ppm of Au also enhanced MSC proliferation while suppressing ROS generation, protein expression of Matrix Metalloproteinase 9 (MMP-9) and endothelial Nitric Oxide Synthase (eNOS), which are associated with MSC migration [24]. Additionally, after collagen was combined with 43.5 ppm of Au nanoparticles (Col–Au), the distinct sheet of fibrils significantly influenced the biological functions of MSCs. Col–Au nanocomposites also demonstrated better biocompatibility, stimulated the expression level of both α5β3 integrin/CXCR4 and Matrix Metalloproteinase 2 (MMP-2), and induced the expression of the endothelialization marker CD31 in MSCs [25]. Thus, Au nanoparticles at an appropriate amount could change the nano-topography and mechanical properties of the polymers or biomolecules to both help facilitate biological function and strengthen differentiation capacity. 

Cell source is one of the main factors involved in tissue regeneration. In a previous study, ECs layered on vascular grafts were prepared, where they demonstrated superior hemocompatibility and differentiation capacity [10], although the source of the endothelial cells was insufficient. Then, the multipotent MSCs were suggested as a promising tool for regenerative medicine owing to their self-renewal ability, isolation easiness, immunoregulation, and differentiation into various types of cells such as osteoblasts, chondrocytes, and adipocytes [26]. Furthermore, a previous research study also verified that human bone marrow-derived mesenchymal stem cells could be induced to differentiate themselves into endothelial-like cells, where the expression of endothelial markers such as KDR and FLT-1 were observed to significantly increase, similar to how the von Willebrand Factor (vWF) is related to angiogenesis [27]. MSCs also secrete multiple growth factors and cytokines, which are associated with MSC-mediated wound healing, such as how Vascular Endothelial Growth Factor (VEGF) is related to angiogenesis, basic Fibroblast Growth Factor (bFGF) is related to fibroblast migration, and Chemokine Ligand 1 (CXCL1) is related to epithelial cell migration [28]. Previous literature has demonstrated that a PEGylated fibrin patch could induce bone marrow MSCs to become differentiated phenotypes. It was observed that endothelial genes such as VEGF, vWF, and CD31 were significantly expressed in MSCs [29]. 

In the current study, we prepared PEG incorporated with 43.5 ppm of Au nanoparticles (PEG–Au 43.5 ppm) and compared it with pure PEG polymer, hypothesizing that the combination of PEG and 43.5 ppm of Au may improve both biocompatibility and the endothelial differentiation ability of MSCs through in vitro and in vivo assessments. Additionally, we were hoping to evaluate the potent effects of PEG–Au nanocomposites as a simple surface modification method for vascular regeneration engineering. 

## 2. Materials and Methods

### 2.1. Material Preparation

#### 2.1.1. Preparation of Polyethylene Glycol (PEG)

PEG was obtained from Sigma-Aldrich, Burlington, MA, USA (500 mM, MW = 200 kDa). The PEG solution was first diluted 25 times with ddH_2_O to achieve a final concentration of 20 μM (1 mL of PEG and 24 mL of ddH_2_O). Next, the PEG solution was coated onto 10 cm culture dishes and 6, 24, and 96-well plates for 30 min to assure contact with the surface of the culture plates. Then, the residual solution was removed to acquire the nano thin film so it could be applied during the subsequent experiments. 

#### 2.1.2. Preparation of Polyethylene Glycol–Gold Nanoparticles (PEG–Au)

The solution of gold nanoparticles was obtained from Gold Nanotech Inc. (Taipei, Taiwan), while the gold nanoparticles were collected as described in a previous study [30]. Gold NanoTech Inc. utilizes unique and patented technology to physically break down bulk gold into nanoparticles, which is followed by epitaxially stacking these nanoparticles into stacked materials of controlled diameter within the nanometric range. Gold nanoparticles produced by this manufacturing procedure possess distinctive physical properties owing to a special ionic charge that maintains their structure and is different from commercially available nanogold produced by chemical reduction methods. The Au nanoparticles were dispersed into distilled water (approximately 50 ppm), with the diameter of the Au nanoparticles being 3–5 nm. The PEG solution was first mixed with 43.5 ppm of Au nanoparticles and then sonicated for 30 min before being coated onto culture dishes and plates for 30 min. Then, the residual solution was removed to obtain PEG–Au thin films. 

### 2.2. Material Characterization

The UV-Vis spectra of pure PEG, pure Au, and PEG–Au 43.5 ppm were measured by a Helios Zeta spectrophotometer (ThermoFisher, Pittsfield, MA, USA), with the wave ranging from 200 to 800 nm. The typical peak of 520 nm was for the Au nanoparticles. The measurement was described in our previous report [23]. Origin Pro 8 (Originlab Corporation, Northampton, MA, USA) software was used to analyze and quantify the data. 

Transmission Fourier transform infrared (FTIR) spectra were obtained through an FTIR spectrometer (Shimadzu Pretige-21, Kyoto, Japan). A solution containing 0.06 g potassium bromide (KBr, Sigma-Aldrich, Burlington, MA, USA) was mixed with 200 mg of pure PEG and PEG–Au 43.5 ppm. The samples were scanned eight times in the scanning range of 500–4000 cm^−1^ with a 2 cm^−1^ resolution to obtain the spectrum [31]. 

The surface morphology of the pure PEG and PEG–Au 43.5 ppm dry coating was observed through a MFP-3D atomic force microscope (Asylum Research, Santa Barbara, CA, USA). A silicon cantilever with a 2.0 N/m condition was used to obtain topography images, which were captured in AC mode at 512 × 512 pixels. 

### 2.3. Free Radical Scavenging Assay

The free radical scavenging ability of the pure PEG and PEG–Au 43.5 ppm nanocomposites were measured using DPPH assay (2,2-diphenyl-1-picrylhydrazyl) (Sigma-Aldrich, Burlington, MA, USA), which was described in a previous study [32]. The nanocomposites were first dissolved in deionized water. Next, 1 mL of the nanocomposites was mixed with DPPH at ratio of 1:3 and then incubated in the dark at room temperature for 90 min. The absorbance of each sample was measured at 539 nm using an ultraviolet-visible spectrophotometer (Helios Zeta, Thermo, Waltham, MA, USA). Free radical scavenging ability was calculated based on the formula: % Inhibition ratio = [1 − (absorbance of test sample/absorbance of control)] × 100%. The absorbance value of the test sample and the control were represented as nanomaterials and deionized water, respectively.

### 2.4. Contact Angle Test

The pure PEG and PEG–Au 43.5 ppm nanocomposites were first added on silicon substrates. Next, 0.7 μL of distilled water was dropped on the surface of nanocomposites. The water contact angles of these nanocomposites were determined by PGX model instrument at room temperature. Pure Au nanoparticles were used as the control group.

### 2.5. Cell Culture of Wharton’s Jelly-Derived Mesenchymal Stem Cells 

The MSCs were acquired from the Wharton’s jelly tissue of a human umbilical cord which were kindly gifted by Prof. Woei-Cherng Shyu [33]. The cells were cultured in an H-DMEM medium (Invitrogen), supplemented with 10% FBS (1% (*v/v*) antibiotics 100 U/mL P/S and 1% sodium pyruvate) and removed with 0.05% trypsin–EDTA after reaching confluence (37 °C, 5% CO_2_). Vascular Endothelial Growth Factor (VEGF, 50 ng/mL, Prospec-Tany TechnoGene Ltd., Rehovot, Israel) was used to investigate endothelial differentiation capacity. 

The specific surface antigen of the MSCs was characterized by flow cytometry [34]. The MSCs were incubated with antibodies conjugated with Fluorescein Isothiocyanate (FITC) and Phycoerythrin (PE) using the markers: CD14-FITC, CD45-FITC, CD44-PE, and CD105-PE (BD Pharmingen, San Diego, CA, USA). Isotype controls were demonstrated by PE-conjugated IgG1 and FITC-conjugated IgG1 (BD Pharmingen). FACS software (Becton Dickinson LSR II, Canton, MA, USA) was used to analyze the phenotype of the MSCs. Cells at 8th passages were used in this study. 

### 2.6. Biocompatibility Assay 

#### 2.6.1. Examination of Cell Viability

PEG and PEG–Au solution was coated onto a culture dish to perform the following evaluation. First, 6 × 10^3^ per well of MSCs were cultured in 96-well plates with pure PEG and PEG–Au 43.5 ppm coatings. The MSC alone group represented as the control. At 24, 48, and 72 h, an MTT (3-(4, 5-cimethylthiazol-2-yl)2, 5-diphenyltetrazolium bromide) solution (0.5 mg/mL) was added to each well and incubated for 3 h at 37 °C. Next, a Dimethyl-sulfoxide (DMSO) solution was added for 10 min to dissolve the crystals. The absorbance at 570 nm was detected by a SpectraMax M2 reader (Molecular Devices, San Jose, CA, USA).

#### 2.6.2. Examination of Reactive Oxygen Species (ROS) Generation

Intracellular ROS was detected by DCFH-dA (2, 7-Dichlorofluorescin diacetate) (Sigma-Aldrich, Burlington, MA, USA), which was the oxidation-sensitive fluorescent probe. First, 2 × 10^5^ cells of the MSCs were cultured in 6-well plates with either pure PEG or PEG–Au nanocomposite 43.5 ppm coatings for 48 h of incubation at 37 °C. The MSC alone group was expressed as the control group. After incubation, the MSCs were first washed twice with Phosphate-Buffered Saline (PBS). Next, the cells were stained with 10 nM DCFH-dA for 30 min in the dark at 37 °C. The ROS was investigated through a flow cytometer (Becton Dickinson, Canton, MA, USA) and quantified using Flow Jo 7.6 software (Becton Dickinson, Canton, MA, USA) [25]. 

#### 2.6.3. Monocyte and Platelet Activation Test

The human monocytes were acquired from the whole blood of healthy volunteers according to Percoll protocol (Sigma-Aldrich, Burlington, MA, USA) [35] with IRB approval (CE12164) from the Taichung Veteran Hospital. First, 1 × 10^5^ per well of monocytes were seeded onto 24-well plates with pure PEG and PEG–Au 43.5 ppm coatings and then incubated for 96 h at *37* °C in an RPMI condition medium (10% FBS and 1% (*v/v*) antibiotics (10,000 U/mL penicillin G and 10 mg/mL streptomycin)). The monocyte alone exhibited as the control group. Next, a 0.05% trypsin solution was added to harvest the cells. The morphology of the monocytes and macrophages was observed through a microscope. CD68 (marker of macrophages) was immunostained by the primary anti—CD68 antibody (GeneTex Inc., Irvine, CA, USA). 

First, 2 × 10^6^ per well of platelets were seeded on pure PEG and PEG–Au 43.5 ppm coatings for 24 h of incubation. The platelet alone group exhibited as the control. Next, the platelets were fixed with a 2.5% glutaraldehyde solution for 8 h. After 8 h had passed, the platelets were washed twice with PBS and dehydrated using 30% to 100% alcohol after standing at RT for 10 min. After being dried, the morphology of the platelets was observed by scanning electron microscopy (JEOL JEM-5200, JEOL Ltd., Akishima, Tokyo, Japan).

#### 2.6.4. Cell Morphology and Adhesion Ability

The cell morphology and adhesion ability induced by pure PEG and PEG–Au 43.5 ppm nanocomposites were observed by scanning electron microscopy (JEOL JEM-5200, JEOL Ltd., Akishima, Tokyo, Japan). First, 1 × 10^4^ cells per ml of platelet were cultured for 48 h, fixed with a 2.5% glutaraldehyde solution for 8 h, and dehydrated with 30% to 100% alcohol. The cell morphology was investigated by SEM after being dried. 

### 2.7. Cytoskeleton of MSCs 

Cells at the density of 1 × 10^4^ per well were seeded in a 24-well plate coated with various materials for 24 h incubation. First, MSCs were fixed with 4% paraformaldehyde (PFA) for 10 min, permeated with 0.5% Triton X-100 in PBS, and then reacted for 10 min at room temperature (RT). Next, phalloidin (≈6 μM) (Sigma, Burlington, MA, USA) was treated for 60 min in dark at RT. Ultimately, 4, 6-diamidion-2-phenylindole (DAPI) (Invitrogen, White Plains, NY, USA) solution was used to stain the cell nucleus in the dark for 10 min. The cytoskeletal of MSCs were observed by a fluorescent microscope [25].

### 2.8. Assessment of MSCs Migration

The procedure of cell migration assay was followed in a previous study [25]. First, 1 × 10^4^ per well of MSCs were added into Oris seeding stoppers for 24 h of incubation to reach confluency. Next, the stoppers were removed and used as a pre-migration reference (t = 0 h). The post migration was represented at 24 h. Next, 200 μL of a Calcein AM solution (2 μM, Sigma-Aldrich, Burlington, MA, USA) was added to the culture plates and stained for 30 min at the time points of 0 and 24 h. The cells were observed by a Zeiss Axio Imager A1 fluorescence microscope (White Plains, NY, USA). Cell migration distance was semi-quantified using Image J 5.0 software (Becton Dickinson, Canton, MA, USA). The MSC alone represented as the control.

### 2.9. Immunofluorescence Staining

The pure PEG and PEG–Au 43.5 ppm nanocomposites were coated on 15 mm coverslip glasses and placed onto 24 well culture plates. First, 2 × 10^4^ per well of MSCs were cultured on these coverslip glasses and incubated in a conditioned medium. The MSC alone exhibited as the control group. The MSCs were incubated with a 1:300 dilution of primary antibodies, including vinculin and CD31 (Santa Cruz, TX, USA). Next, the coverslip glasses were washed and incubated with a 1:300 dilution of FITC-conjugated secondary antibodies (Santa Cruz, TX, USA) for 1 h. The cell nuclei were detected by a DAPI solution (Invitrogen, White Plains, NY, USA). Before being sealed for subsequent observation, each sample was cautiously washed using a 50% glycerol/PBS solution and placed on slides. The images were captured in the dark by a fluorescence microscope. The fluorescent positive cells were detected and quantified using Image J 5.0 software (Media Cybernetics, Burlington, MA, USA). 

### 2.10. Real-Time PCR Assay

The mRNA in the MSCs was extracted by Trizo1 (lnvitrogen, ThermoFisher, Waltham, MA, USA). The procedures were followed based upon the manufacturer’s instructions. First, 1 × 10^5^ per well of MSCs were cultured in 10 cm culture dishes with pure PEG or PEG–Au 43.5 ppm coatings for 7 days. The MSC alone presented as the control group. Next, the cells were treated with 1 mL of Trizol for 5 min. The MSCs underwent lyolysis by adding 200 μL of chloroform (Sigma-Aldrich, Burlington, MA, USA) for 15 s, standing for 3 min at RT, and then being centrifuged at 12,000 rpm/4 °C for 15 min. Next, the supernatant was removed, and 500 μL of 4 °C isopropanol was added for 10 min of incubation. Next, all of the samples were centrifuged at 12,000 rpm/4 °C for 15 min. Afterwards, the supernatant was removed and washed twice with 1 mL of 75% alcohol. After drying the RNA samples, 20 μL of DEPC-treated water was added to each sample and then quantified by the absorbance at 260 nm using a SpectraMax M2 ELISA reader (Molecular Devices, San Jose, CA, USA). A RevertAidTM First Strand cDNA DNA Synthesis Kit (Fermentas, Canada) was used to further process the synthesis of cDNA. First, 2 μL of oligo (dT) 18 primer and random hexamers (1:1) were added to the RNA sample, with the samples then placed into a gradient polymerase reaction temperature controller at 65 °C for 5 min. Before the samples were reacted at 42 °C for 60 min, 4 μL of 5× reaction buffer, 1 μL of LockTM RNase inhibitor (20 U/mL), 2 μL of dNTP Mix (10 mM), and 1 μL of RevertAidTM M-MuLV Reverse Transcriptase (200 U/mL) were added to each sample. Ultimately, all of the samples were carried out at 70 °C for 5 min to obtain the cDNA. The polymerase chain reaction was processed by a 1Q2 Fast qPCR System using the cDNA as a template with a 10 μL reaction volume based on the manufacturer’s procedures. Then, 0.5 μL of primer (0.3 μM) and 5 μL of enzyme were added to the cDNA sample, with the RNA expression analyzed by the Step OneTM Plus Real-Time PCR System.

### 2.11. Rat Subcutaneous Implantation

Female Sprague–Dawley (SD) rats (age: 2–3 months, weight: 300–350 g) were obtained from National Laboratory Animal Center (Taipei, Taiwan), which were used after receiving approval from the Animal Care and Use Committee (La-1071565). In the in vivo experiments, PEG and PEG–Au composite solutions were coated on a glass coverslip (15 mm) and implanted to rat subcutaneous tissue. After being given local anesthesia, the nanocomposites were subcutaneously implanted into a 10 mm^2^ incised dorsal skin area of rats for 4 weeks. Then, the rats (n = 5) were sacrificed, and the tissues with implanted materials were further subjected to histological investigations. The formation of fibrous capsule in six sites was investigated by Hematoxylin and Eosin (H&E) staining, and the thickness of encapsulated fibrotic tissues was calculated and quantified by Image J 5.0 software. The marker of endothelialization, CD31, was also investigated in tissue [36]. Moreover, TUNEL assay was applied to detect apoptotic cells in the rat models. During the late stages of cell apoptosis, DNA fragmentations could be targeted by labeling 3′-hydroxyl termini. An In Situ Cell Death Detection Kit, AP (Roche Diagnostics, USA) was purchased to detect the apoptotic cells and put to use by following the protocol provided by the manufacturer. The Olympus ix71 fluorescence microscope (Tokyo, Japan) was applied to further analyze the fluorescence intensity. DAPI (4′,6-diamidino-2-phenylindole) staining was used to target the nuclear DNA in fixed cells. The MSC alone represented as the control. All results were represented as mean ± SD.

### 2.12. Statistical Analysis

Samples for each experiment (n = 3–6) were obtained, with the results represented as mean ± Standard Deviation (SD). To avoid uncertainty, all of the experiments were independently processed in triplicate. Single-factor Analysis of Variance (ANOVA) and Student’s t-test were used to evaluate the statistical difference between the different groups. The Bonferroni method was used for post hoc analysis in ANOVA. A *p* value of less than 0.05 was statistically significant.

## 3. Results

### 3.1. Characterization of PEG–Au Nanocomposites

First, a UV-Vis spectroscopy was applied to confirm the presence of Au nanoparticles on PEG. As shown in Figure 1A, the absorption wavelength figured out a typical peak at 520 nm from Au nanoparticles and PEG–Au 43.5 ppm groups. Next, the FTIR spectra demonstrated the functional groups in Au nanoparticles, pure PEG, and PEG incorporated with 43.5 ppm of Au nanoparticles (Figure 1B). The specific peaks of pure PEG were at 2902.59 cm^−1^ (-CH_2_ stretching), 1645.84 cm^−1^ (-CH_2_ scissoring), and 1097.33 cm^−1^ (C–O–C stretching). These peaks were also found in the PEG–Au 43.5 ppm group. However, the specific peaks of Au nanoparticles were at 3430.72 cm^−1^. In the PEG–Au 43.5 ppm group, the peaks of CH_2_ stretching, CH_2_ scissoring, and C-O-C stretching was shifted to 2902.75 cm^−1^, 1646.32 cm^−1^, and 1097.63, respectively. The above evidence indicated the modification of Au nanoparticles on PEG nanofilm. Furthermore, the surface morphology of pure PEG and PEG combined with 43.5 ppm of Au nanoparticles was observed by AFM. The images exhibited that the pure PEG was homogenous and uniform. However, the surface morphology became strip-like, while Au nanoparticles were decorated onto PEG (Figure 1C). The Reactive Oxygen Species (ROS) scavenging ability of pure PEG and the PEG–Au 43.5 ppm nanocomposites was also evaluated. In Figure 1D, the semi-quantitative results demonstrate that PEG containing 43.5 ppm of Au nanoparticles had better capture ability (≈6.8 fold, *p* < 0.01), and pure PEG had the value of ≈2.5 fold (*p* < 0.01) compared to the control, respectively. The hydrophilicity property of biomaterials plays an important role for cell attachment. Figure 1E shows that the water on PEG–Au 43.5 ppm had the lowest water contact angle (18.5°, *p* < 0.01), followed by pure PEG (26°, *p* < 0.05) as compared to the control (28.8°) group, indicating that the modification of Au nanoparticles onto PEG polymer makes the nanocomposites become more hydrophilic for cell adhesion.

### 3.2. Biocompatibility Assessments of MSCs Culturing on PEG–Au Nanocomposites

The phenotypes of the MSCs were characterized through detecting specific surface antigens using flow cytometry. The negative markers CD14 (1.64%) and CD45 (0.8%) were highly expressed in hematopoietic cells and immune cells, respectively. The specific antigens CD44 (98.6%) and CD105 (99.7%) for the MSCs were significantly detected (Figure 2A). Thus, the MSCs were used for the subsequent assessments.

The cell viability between different materials was investigated by MTT assay. As seen in Figure 2B, cell viability significantly increased in the PEG–Au 43.5 ppm group at 48 and 72 h (OD_570nm_ = 1.6 and 1.9, *p* < 0.01), followed by the pure PEG group (OD_570nm_ = 1.45 and 1.62, *p* < 0.01) when compared to the control group. Moreover, the intracellular ROS generation of MSCs induced by different materials was investigated at 48 h. Figure 2C demonstrates the semi-quantitative result of ROS production being the lowest in the PEG–Au 43.5 ppm group (≈0.55 fold, *p* < 0.01), which was followed by the pure PEG group (≈0.8 fold, *p* < 0.05) when compared to the control group. The above results indicate that PEG–Au 43.5 ppm could strengthen cell proliferation and anti-ROS generation ability. 

When inflammatory response occurs, the activation of platelets and monocytes can be clearly observed. The degree of platelet activation induced by pure PEG and PEG–Au 43.5 ppm was observed, with the SEM images at 96 h displayed in Figure 3A. Based upon the SEM images, the non-activated platelets (round morphology) were mostly discovered in the PEG–Au 43.5 ppm group. On the contrary, activated platelets (flattened morphology) were found in the pure PEG and control groups. Additionally, the platelet adhesion in each group was also semi-quantified, as shown in Figure 3C. The results elucidate that the lowest adhesion occurred in the PEG–Au 43.5 ppm group (≈0.22 fold, *p* < 0.01), which was followed by the pure PEG group (≈0.92 fold, *p* < 0.05) compared to the control (one-fold). The expression of macrophage marker CD68 was investigated at 96 h through immunofluorescence staining (Figure 3B). The quantitative results based on fluorescence intensity demonstrated that the PEG–Au 43.5 ppm group induced the lowest expression of CD68 (≈0.3 fold, *p* < 0.01), which was followed by pure PEG group (≈0.9 fold, *p* < 0.05) when compared to the control group (Figure 3D). The conversion ratio of monocyte (≈5 µm) to macrophage (≈40 to 45 µm) in the pure PEG and PEG–Au 43.5 ppm groups at 96 h is displayed in Figure 3E. The conversion ratio stimulated by PEG–Au 43.5 ppm was the lowest (≈0.3 fold, *p* < 0.01), followed by pure PEG (≈0.9 fold, *p* < 0.05) when compared to the control group. The above results indicate that PEG–Au 43.5 ppm exhibited excellent anti-inflammatory capacities. 

### 3.3. MSCs Migration Induced by PEG–Au Nanocomposites

The length of F-actin and the expression of vinculin were both investigated after 24 h of incubation through both phalloidin staining and immunofluorescent staining assay, with the results demonstrated in Figure 4A. Indeed, the F-actin length of MSCs induced by PEG–Au 43.5 ppm was the greatest (≈1.65 fold, *p* < 0.01), followed by the pure PEG group (≈1.08 fold, *p* < 0.05) when compared to the control group (Figure 4B). The expression of vinculin in MSCs was also the highest in the PEG–Au 43.5 ppm group (≈1.98 fold, *p* < 0.01), followed by the pure PEG group (≈1.28 fold, *p* < 0.05) (Figure 4C). Furthermore, MSC migration influenced by different materials was evaluated using Calcein–AM staining, with the real-time images at 0 and 24 h displayed as Figure 4D. The distance that MSCs migrated into the gap zone was significantly induced by PEG–Au 43.5 ppm (45.4 μm, *p* < 0.01), followed by pure PEG (25.36 μm, *p* < 0.05) when compared to the control group (20.78 μm) (Figure 4E). The above results conclude that PEG–Au 43.5 ppm could effectively enhance both cell attachment and migration ability. 

### 3.4. Endothelial Differentiation Capacity Promoted by PEG–Au Nanocomposites on Day 7

To investigate the endothelialization induced by different materials in MSCs, the expression of CD31 was measured by immunofluorescence staining and real-time PCR. The phenotypes of endothelia were observed at Day 7, with the fluorescence images displayed as Figure 5A. Based upon the quantitative results of real-time PCR, CD31 mRNA expression induced by PEG–Au 43.5 ppm was the greatest (≈1.42 fold, *p* < 0.01), followed by pure PEG (≈1.18 fold, *p* < 0.01) when compared to the control group (1 fold) (Figure 5B). The semi-quantitative data based on fluorescence intensity also demonstrated that the highest expression of CD31 was seen in the PEG–Au 43.5 pm group (≈1.68 fold, *p* < 0.01), followed by pure PEG (≈1.3 fold, *p* < 0.01) (Figure 5C). The evidence elucidates that PEG–Au 43.5 ppm could facilitate endothelialization for MSCs.

### 3.5. Assessments of In Vivo Implantation for PEG–Au Nanocomposites

The nanomaterials, pure PEG and PEG–Au 43.5 ppm, were subcutaneously implanted into our rat models for 4 weeks to evaluate the in vivo biocompatibility. The anti-inflammatory effect and endothelial differentiation capacity were further investigated after 4 weeks of implantation. In Figure 6A, the effect of each nanomaterial on fibrotic encapsulation in tissues were analyzed through H&E staining. The results are semi-quantified in Figure 6C, indicating that PEG–Au 43.5 ppm induces the lowest formation of fibrous capsule (≈0.72 fold, *p* < 0.01), followed by pure PEG (≈0.9 fold, *p* < 0.05) when compared to the control group (one-fold). Additionally, the expression of endothelial marker CD31 in tissue was also investigated through IHC staining, with the images displayed in Figure 6B. Based upon the expression intensity, the semi-quantitative results demonstrate that CD31 expression level was the highest in the PEG–Au 43.5 ppm group followed by pure PEG (≈1.12 fold, *p* < 0.01) and the control group (one-fold) (Figure 6D). The above results elucidated that PEG–Au 43.5 ppm significantly attenuated foreign body responses as well as induced endothelialization in vivo assessments for tissue regeneration.

Moreover, TUNEL assay was applied to target the late stage of apoptotic cells (DNA fragmentation) for biosafety concerns, with the images demonstrated in Figure 7A. The semi-quantification of TUNEL-positive cells showed there was no significant difference between the treated groups (PEG: ≈1.02-fold, PEG–Au 43.5: ≈0.98-fold) and control group (one-fold) (Figure 7B). The above result indicated that both pure PEG and PEG–Au 43.5 ppm would not harm the surrounding tissues after implantation.

Current research has validated that PEG–Au 43.5 ppm nanocomposites possess better biocompatibility, biological functions, and greater endothelial differentiation capacity. In vivo assessments have also evaluated that PEG–Au 43.5 ppm exhibited a strong ability regarding anti-inflammatory capacity and induction of endothelialization, as well as the lowest injury rates for normal tissue.

## 4. Discussion

Blood vessels are highly complicated in structure that lacks regeneration capacity. Various functions of the blood vessels are exhibited through the presence of the Extracellular Matrix (ECM). The different composition, thickness, and structure of ECM support the formation of blood vessels such as arteries, veins, and capillaries [37]. However, due to lack of ECM which can provide support for cell proliferation, migration, and attachment, creating vascular conduits to mimic native blood vessels is difficult. ECM is also a major component in the process of angiogenesis [38]. Vascular replacement or revascularization are the standard procedures for clinical surgeries. However, autologous grafts are commonly sufficient due to limited vessel availability and difficult to be acquired. Thus, the demand for appropriate vascular grafts with long-term patency has gradually increased [39]. Surface modification through chemical, physical, and biomolecule immobilization have been well investigated to improve the biocompatibility of polymeric grafts. Previous literature has also verified that modifying the surface properties of biomaterials at the nano-scale level can enhance cell adhesion capacity [40]. For instance, the surface of PLGA film was formulated into spherical 200 nm features, which could facilitate endothelial cell adhesion as compared to smooth PLGA and 100 nm or 500 nm PLGA surface features [41]. In the current study, the homogenous feature of pure PEG was observed to become strip-like after incorporating with 43.5 ppm Au nanoparticles. Furthermore, hydrophilic measurement also indicated that PEG–Au 43.5 ppm could enhance cell adhesion capacity.

As mentioned above, the nano-scale surface modified with nanoparticles under an appropriate condition could provide focal points for filopodia of cells to attach more efficiency [42]. Therefore, the noble metals such as Au and silver (Ag) nanoparticles have attracted the attention due to their special characteristics. For instance, Ag nanoparticles have verified to be a potential anti-microbial nanomaterial [43]. Our research team have developed the nanocomposites fabricating Ag nanoparticles with Col, which demonstrated the longer stress fiber extension for MSCs adhesion and migration [44]. Additionally, Au nanoparticles were proved to be non-cytotoxic, which owned superior biocompatibility for biomedical applications such as vascular regeneration [45]. A piece of literature also described that the Col–Au nanomaterial significantly stimulated MSCs to have more protrusions (filopodia) and elongation for attachment [25]. In line with current results, the PEG nanofilm modified with 43.5 ppm of Au nanoparticles was observed to promote the expression of vinculin and elongate F-actin fiber, supporting that PEG–Au 43.5 ppm could significantly influence cell morphology to promote MSCs attachment.

The literature indicated Au functionalized nanoparticles had an efficient radical scavenging property [46], which was corresponded to the data for PEG–Au 43.5 ppm nanocomposites (Figure 1D). Furthermore, in line with our results, after 43.5 ppm of Au nanoparticles were decorated on the synthetic polymer PEG, the activation of platelets and monocytes, as well as the ROS production, significantly decreased (Figure 3). The interactions between monocytes and platelets could enhance monocyte migration into blood vessel walls, causing atherogenesis [47]. Thus, the appropriate addition of Au nanoparticles can remarkably facilitate cell viability and anti-oxidative capacity for MSCs. As mentioned above, MSCs could provide various functions such as secreting cytokines for tissue repair. MSCs can also be induced to differentiate into EC-like cells [27].

Promoting angiogenesis and inhibiting thrombosis are major factors in vascular regeneration. Recent research study created a vascular graft composed of polycaprolactone-methoxy polyethylene glycol as the inner layer and poly (4,4′-methylenebis (phenyl isocyanate)-alt-1,4-butanediol/di(propylene glycol)/polycaprolactone) (PU-PCL) as the outer layer. Focusing on the inner layer of the graft, polycaprolactone-methoxy polyethylene glycol significantly improved the hydrophilicity and anti-thrombogenic properties when compared to PCL or PU-PCL membranes. Additionally, based on in vivo assessment, the polycaprolactone-methoxy polyethylene glycol membrane strengthened the differentiation of rat bone marrow stem cells into ECs and also significantly facilitated the protein and mRNA gene expression of CD31, Flk-1, and vWF, which was associated with angiogenesis [48]. Our previous study also demonstrated that Col fabricated with 43.5 ppm of Au nanoparticles significantly enhanced CD31 expression as well as promoted the endothelial differentiation for MSCs [25], which was corresponded to Figure 5 in current research.

The biomechanical and hemodynamic properties of vascular grafts are highly concerned. The surface of Col [31] and FN [49] nanofilm modified with different concentration (17.4, 43.5, and 174 ppm) of Au nanoparticles was discussed in our previous studies. The Young’s modulus analysis demonstrated that the elasticity of Col–Au and FN–Au nanocomposites decreased with the increased concentration of Au. Furthermore, the surface roughness observed through AFM evaluated the thicker fibrils due to more collagen aggregation caused by 43.5 ppm of Au nanoparticles. The measurement of hydrophilicity property demonstrated that the increased concentration of Au lead to better wettability for Col–Au and FN–Au nanocomposites. Additionally, the thermogravimetric analysis (TGA) figured out that the pyrolytic temperatures were increased with the addition of 43.5 ppm Au nanoparticles into Col and FN. Based on the above evidence, we inference that the PEG nano film modified with the optimal concentration of Au (43.5 ppm) will improve mechanical resistance to the blood flow.

Above all, our findings indicate that the modification of 43.5 ppm Au nanoparticles on PEG nanofilm leads to better surface topography, thus promoting cell attachment. The overloading of Au nanoparticles caused particle aggregation in the polymer matrix, which was verified in our previous research [23,50]. Furthermore, MSC migration ability and endothelialization capacity were remarkably induced by PEG–Au 43.5 ppm when compared to pure PEG. In vivo assessments also declared the excellent anti-inflammatory ability and differentiation ability promoted by PEG–Au 43.5 ppm. The evidence validated that the PEG–Au 43.5 ppm nanocomposite coating could improve both the hemocompatibility and cell behavior of MSCs for the development of clinical blood contacting devices.

## 5. Conclusions

In the present study, we prepared pure PEG and PEG incorporated with 43.5 ppm of Au nanoparticles in order to analyze their biocompatibility and endothelial differentiation capacity through in vitro and in vivo assessments. The results have first demonstrated that PEG–Au containing 43.5 ppm of Au nanoparticles significantly strengthened the cell viability and anti-oxidative ability in MSCs. Meanwhile, the activation of platelets as well as monocyte–macrophage conversion was well inhibited by PEG–Au 43.5 ppm, indicating excellent biocompatibility for the nanomaterials. Furthermore, PEG–Au 43.5 ppm enhanced migration ability and also facilitated the endothelialization capacity of MSCs. Additionally, in vivo assessments elucidated lower capsule formation and better enhancement of endothelialization, which were induced by PEG containing 43.5 ppm of Au nanoparticles. Moreover, TUNEL assay also demonstrated that PEG–Au 43.5 ppm was a biosafety nanomaterial that would not significantly induce cell apoptosis when implanted into our animal model. Based upon the above evidence, PEG–Au 43.5 ppm exhibited better biocompatibility and induction of endothelialization, suggesting that PEG incorporating 43.5 ppm of Au nanoparticles can be a potential nanomaterial for vascular regeneration engineering.

## Figures and Tables

**Figure 1 polymers-13-04265-f001:**
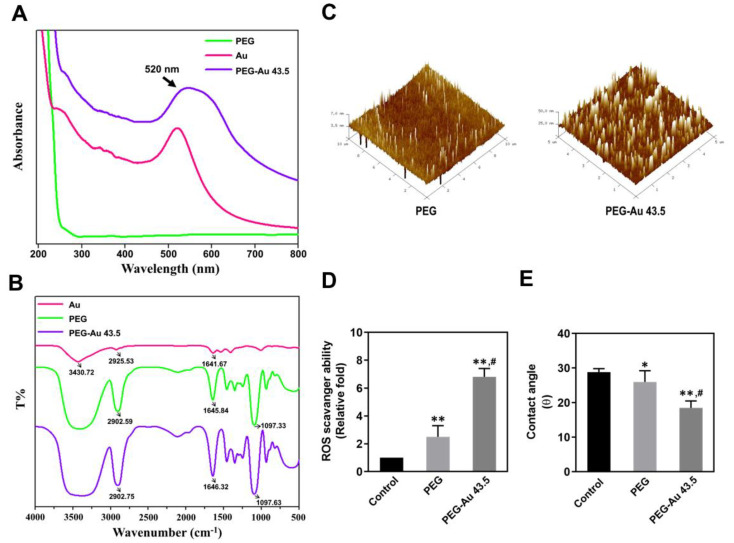
Material characterization. (**A**) The UV-Vis spectra of pure PEG, pure Au, and PEG–Au 43.5 ppm. The absorbance peak at 520 nm evaluated the presence of Au nanoparticles for pure Au and PEG combined with 43.5 ppm of Au. (**B**) The FTIR spectra of Au nanoparticles, pure PEG, and PEG–Au 43.5 ppm. (**C**) The AFM topography of 3D images for pure PEG and PEG–Au 43.5 ppm nanocomposites. (**D**) The free radical scavenging ability of pure PEG and PEG–Au 43.5 ppm. The ROS scavenging ability of pure PEG is slightly better than the control. However, the PEG–Au 43.5 ppm group demonstrated the greatest scavenging ability. Deionized water represented as the control group. (**E**) The hydrophilicity property of PEG and PEG–Au 43.5 ppm was investigated. The semi-quantitative results of the average water contact angle (θ) evaluated that PEG–Au 43.5 ppm was the lowest. The contact angle without water was θ = 0°. Pure Au nanoparticles were represented as the control group. Data are represented as mean ± SD (n = 3). * *p* < 0.05, ** *p* < 0.01: compared to the control group. # *p* < 0.05: compared to the pure PEG group.

**Figure 2 polymers-13-04265-f002:**
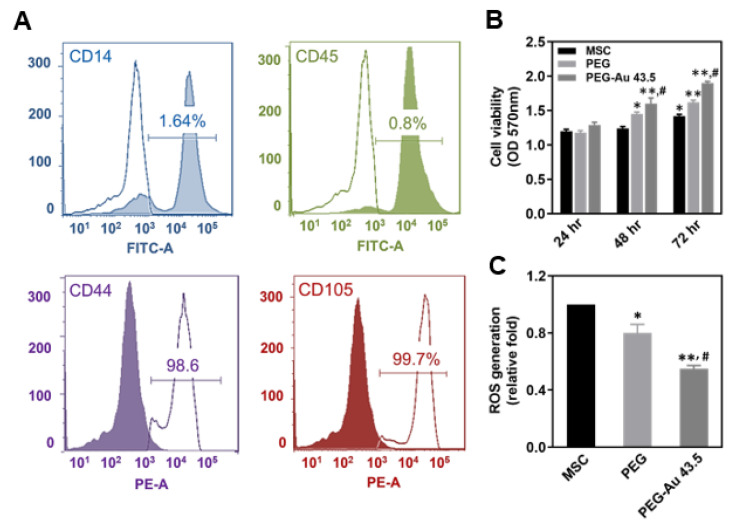
Phenotype characterization and biocompatibility assessments of MSCs. (**A**) The antibodies were first conjugated with Fluorescein Isothiocyanate (FITC) and Phycoerythrin (PE), with the following markers being CD14-FITC, CD45-FITC, CD44-PE, and CD105-PE. Next, the surface antigens of MSCs were analyzed by flow cytometry. (**B**) The cell viability of MSCs after culturing on pure PEG and PEG–Au 43.5 ppm at various time periods (24, 48, 72 h) were investigated by MTT assay. The cell viability was greatest at 72 h in the PEG–Au 43.5 ppm group compared to the control group. Data are expressed as mean ± SD (n = 6). * *p* < 0.05, ** *p* < 0.01: compared to the MSC alone group. # *p* < 0.05: compared to the pure PEG group. All the results are representative of one of six independent experiments. (**C**) The intracellular ROS generation of MSCs culturing on pure PEG and PEG–Au 43.5 ppm was detected through 2,7-dichlorofluorescein diacetate (DCFH-dA) and analyzed by flow cytometry. The results were semi-quantified. Data are represented as mean ± SD (n = 3). * *p* < 0.05, ** *p* < 0.01: compared to the MSC alone group. # *p* < 0.05: compared to the pure PEG group. All the results are representative of one of three independent experiments.

**Figure 3 polymers-13-04265-f003:**
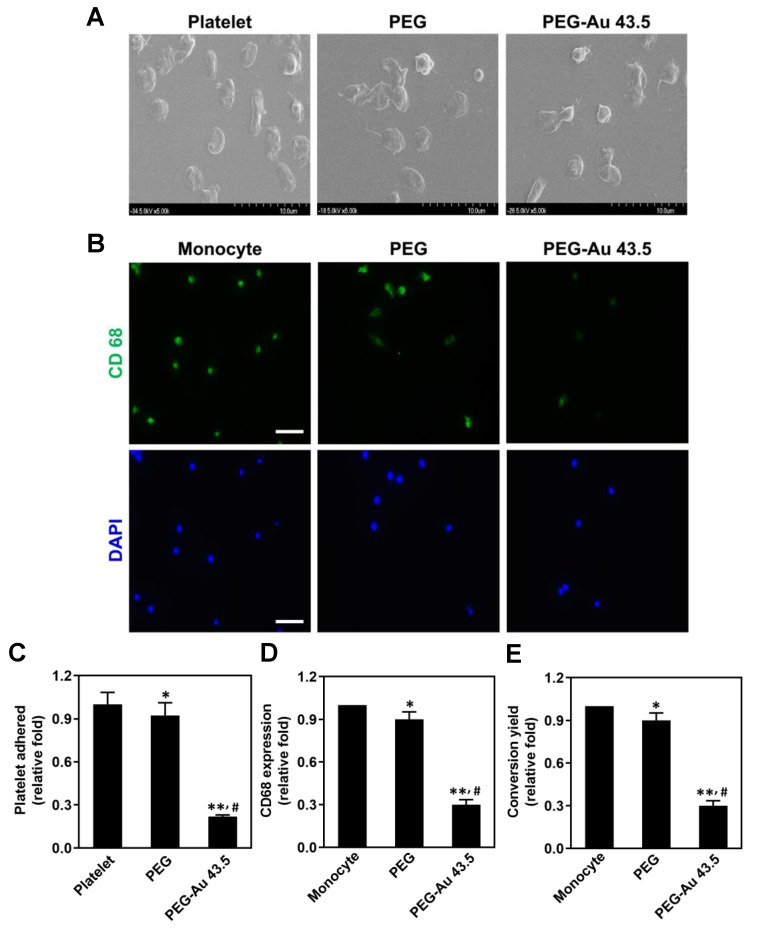
Assessments of platelet activation and monocyte conversion induced by pure PEG and PEG containing 43.5 ppm of Au at 96 h. (**A**) Images of SEM demonstrated the adhesion and activation of human blood platelets in different materials. (**B**) The expression of CD68 for macrophages on various materials at 96 h. The cells were immunostained by primary anti-CD68 antibodies and conjugated with FITC—immunoglobin secondary antibodies (green color). Cell nuclei were stained by a DAPI solution (blue color). Scale bar = 20 μm. (**C**) The data quantified from the degree of activation score indicated that PEG–Au 43.5 ppm could inhibit platelets to adhere. (**D**) The semi-quantitative results of CD68 expression based upon fluorescence intensity. (**E**) The monocyte conversion yield demonstrated that PEG–Au 43.5 ppm would not stimulate the activation of monocytes and better attenuate immune response. Data are displayed as mean ± SD (n = 3). * *p* < 0.05, ** *p* < 0.01: compared to the platelet/monocyte alone group. # *p* < 0.05: compared to the pure PEG group.

**Figure 4 polymers-13-04265-f004:**
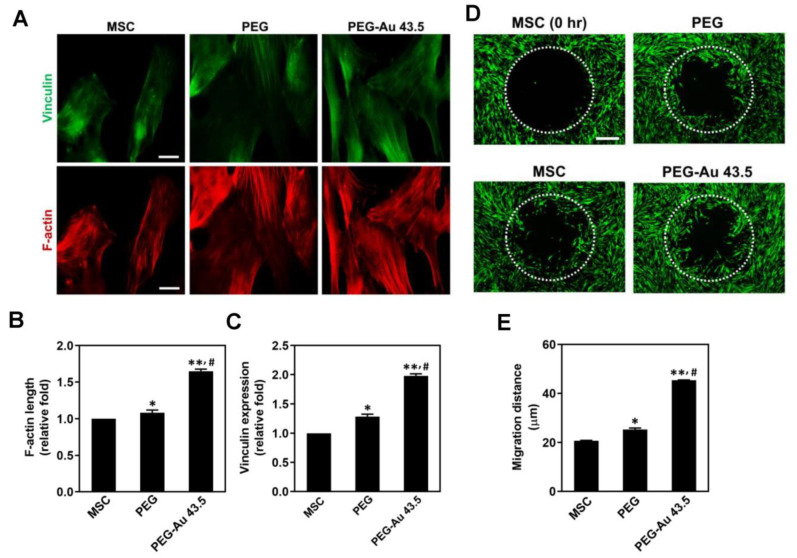
Evaluation of F-actin/vinculin expression and cell migration ability for MSCs incubating with pure PEG and PEG–Au 43.5 ppm at 24 h. (**A**) Firstly, F-actin (red color) was stained using Alexa phalloidin, while vinculin was immunostained using primary anti-vinculin antibodies conjugated with secondary FITC–immunoglobulin antibodies (green color). Cell nuclei were detected by DAPI solution (blue color). Scale bar = 20 μm. The (**B**) length of F-actin and (**C**) expression of vinculin were semi-quantified based upon fluorescence intensity. The results are represented as three independent experiments. Data are presented as mean ± SD. (**D**) The migration ability of MSCs was investigated at 24 h. MSCs migrating into the gap zone area was observed by fluorescence microscopy. The cells were stained by calcein–AM (2 μM) before observation. Scale bar = 200 μm. (**E**) The semi-quantitative results for the migration distance of MSCs at 24 h. Data are represented as mean ± SD (n = 3). * *p* < 0.05, ** *p* < 0.01: compared to the MSC alone group. # *p* < 0.05: compared to the pure PEG group.

**Figure 5 polymers-13-04265-f005:**
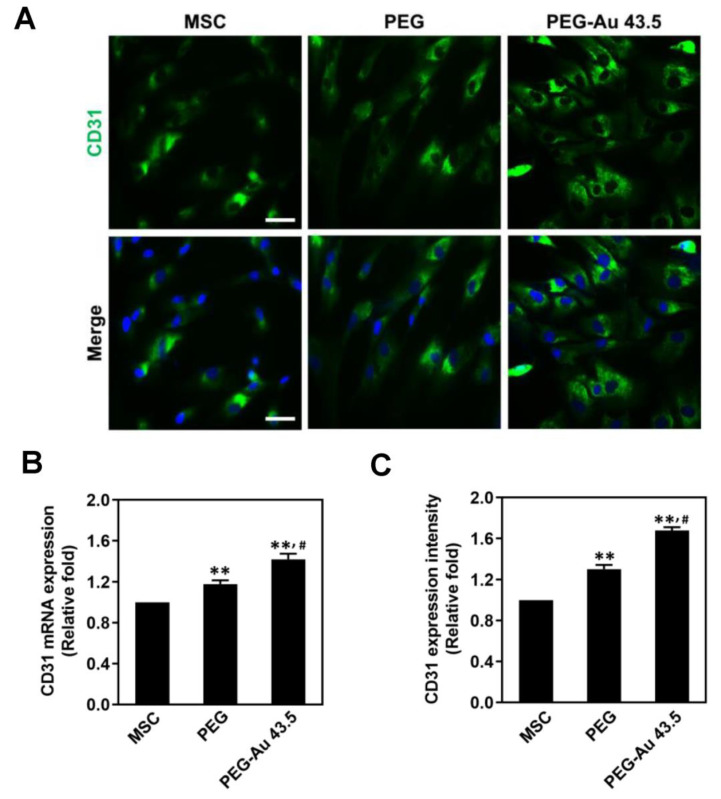
The endothelial differentiation ability of MSCs culturing on pure PEG and PEG–Au 43.5 ppm at Day 7. (**A**) The MSCs were first stained with primary CD31 antibodies which were represented as an endothelial marker, followed by secondary FITC-conjugated immunoglobin antibodies (green color). Cell nuclei were located by DAPI (blue color). The images were merged with DAPI stained cell nuclei. Scale bar = 10 μm. (**B**,**C**) The mRNA expression of CD31 in MSCs was investigated by real-time PCR. The semi-quantitative result evaluated that PEG–Au 43.5 ppm could significantly induce endothelial differentiation. Furthermore, a similar trend was also observed from the quantified data based upon CD31 fluorescence intensity. Data are represented as mean ± SD. ** *p* < 0.01: compared to the MSC alone group. # *p* < 0.05: com-pared to the pure PEG group.

**Figure 6 polymers-13-04265-f006:**
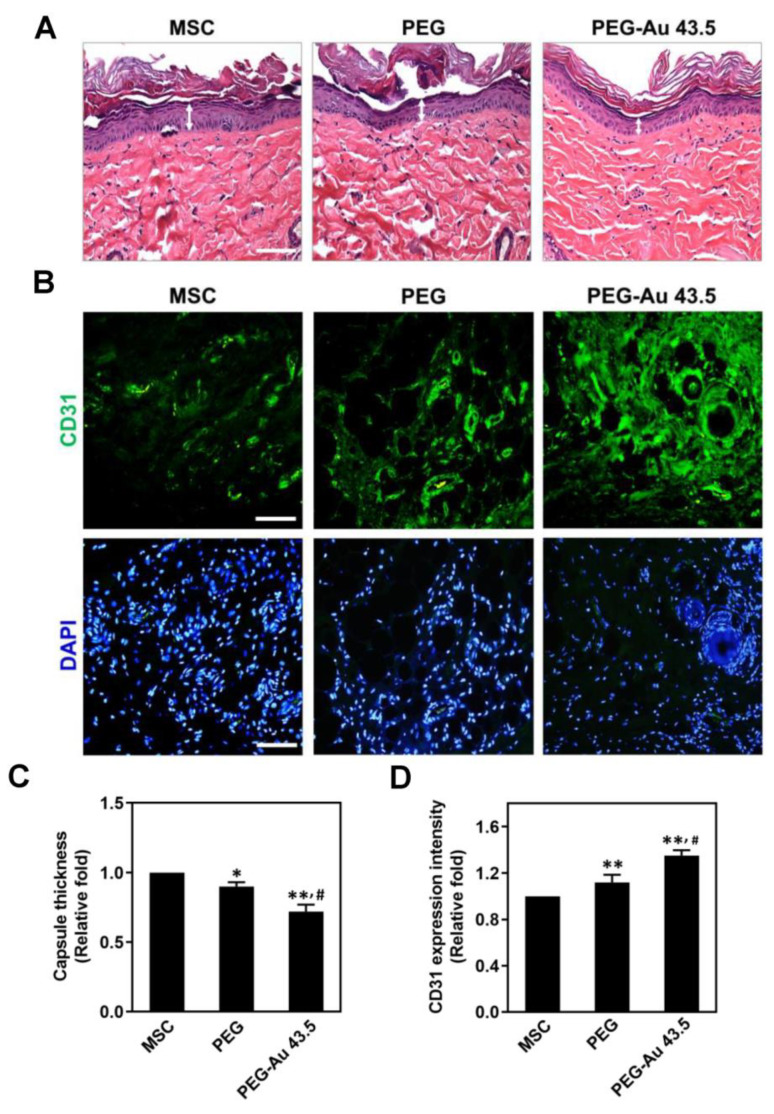
Evaluation of inflammation response and endothelialization capacity after 4 weeks of subcutaneous implantation of pure PEG and PEG–Au 43.5 ppm. (**A**) The capsule formation was investigated by H&E staining. (**B**) The expression of CD31 (green color) in animal tissue was detected by IHC staining. The white arrow indicated the thickness of fibrous capsule. DAPI was used to stain cell nuclei. Scale bars = 50 μm. (**C**) The semi-quantitative result of capsule thickness. The results indicated that PEG–Au 43.5 ppm could effectively decrease capsule formation in tissue, demonstrating its better anti-inflammation ability. (**D**) The semi-quantitative data of CD31 expression intensity indicating PEG–Au 43.5 ppm could significantly induce endothelialization in animal models. The number of rats was 5 (n = 5). Data are represented as mean ± SD. * *p* < 0.05, ** *p* < 0.01: compared to the MSC alone group. # *p* < 0.05: compared to the pure PEG group.

**Figure 7 polymers-13-04265-f007:**
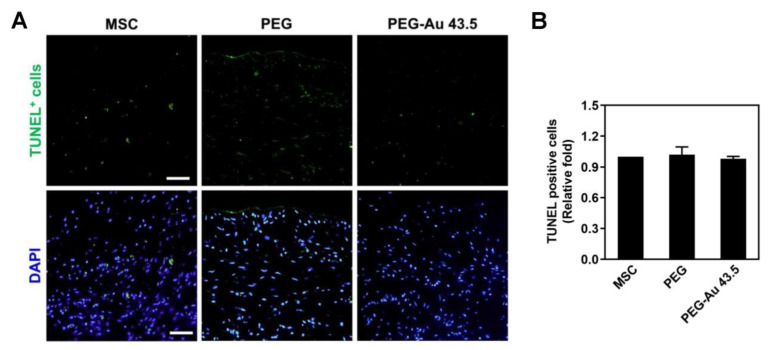
The later stage of cell apoptosis investigated by TUNEL assay after subcutaneous implantation. (**A**) During the later stage of cell apoptosis, DNA will become highly fragmented. It can attach a modified dUTP to the 3′-OH end of the damaged DNA using the enzyme terminal deoxynucleotidyl transferase dUTP nick end labeling (TUNEL, green color) reaction. (**B**) The semi-quantitative result indicates that there is no significant difference between the treated groups and the control group, while also elucidating PEG–Au 43.5 ppm as a biosafety nanomaterial that does not injure the surrounding tissue in rat models. DAPI was used to locate cell nuclei (blue color). The fluorescence was detected by a confocal microscope. Scale bar = 50 μm. The number of rats was 5 (n = 5). Data are presented as mean ± SD.

## Data Availability

Data are contained with the article.

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
