# Peer review of "Evaluation of the Biocompatibility and Endothelial Differentiation Capacity of Mesenchymal Stem Cells by Polyethylene Glycol Nanogold Composites"

_polymers, 2021, doi:10.3390/polym13234265_

Round 1

Reviewer 1 Report

Dear Editor,

The manuscript (polymers-1454277) reports a study on PEG-Au assembly as a potential surface coating for vascular grafts. The study is very similar to a recent publication by the authoring team:

 (https://doi.org/10.3390/cells10112854) and even some results (figure 1, e.g.) have been repeated. Moreover, such a composite system per se is not that novel. Therefore, the authors should elaborate and highlight the importance and novelty of their work. My other major comments include: 

1- The title is vague, what do the authors mean by physical nanogold composites?

2- Title; Polyethylene glycol not glycerol.

3- Abstract, line 22; MSCs have been incorporated into the nanoparticles? This is not the case here.

4- Page 7, line 318; What does decoration mean? Later UV-Vis and FTIR results contradict the hypothesis of surface decoration when the absorbance intensity declines. 

5-FTIR results; I was wondering if no peak shift or disappearance takes place after Au inclusion? If so, how these components interact (bond)?

6- What is the ROS scavenging mechanism?

7- The FTIR spectra once again imply the presence of the Au NPs within the polymer, solely all PEG peaks are shown in both spectra. If so, why water contact angle declines despite entrapment of Au NPs within the polymer.

8- Figure 1; What is the control?

9- Were the samples sterilized? How?

10- The optimum interaction of MSC reflected in their larger F actin filaments could be related to biomechanics of the substrate. PEG-Au is assumed to be stiffer than PEG and this stiffness discrepancy could lead to formation of larger F-actin filaments. How the chemical and mechanical effects could be separately analysed and discussed?

11- Page 9, last paragraph; How surface nanotopography and surface charge play a role in cell-material interaction?

12- Figure 5A, merge with what?

13- Discussion, 1st paragraph; English editing is needed for this part of discussion.

14- Page 15, line 511-523; This paragraph has nothing to do with the current study and does not justify the results related to PEG-Au for surface coating of vascular grafts.

15- line 525-545, What this literature review is to imply? Any relevance to the results to be discussed?

16- Line 546-547; what this sentence means? In general, discussion is indeed poor and vague.

17-line 556, or surface topography? no other change took place, e.g. mechanical stiffness, surface charge, etc.?

Author Response

Reviewer 1

Comments and Suggestions for Authors

The manuscript (polymers-1454277) reports a study on PEG-Au assembly as a potential surface coating for vascular grafts. The study is very similar to a recent publication by the authoring team:

 (https://doi.org/10.3390/cells10112854) and even some results (figure 1, e.g.) have been repeated. Moreover, such a composite system per se is not that novel. Therefore, the authors should elaborate and highlight the importance and novelty of their work. My other major comments include: 

1- The title is vague, what do the authors mean by physical nanogold composites?
(1) The physical gold nanoparticles were purchased from Gold NanoTech, Inc (GNT, Taiwan). The GNT Gold is 99.99% pure, manufactured by physical vapor deposition (PVD) processing, which is different from chemical synthesis used by commercially available nanogold products. GNT Gold contains no other heavy metals or toxic compounds.
(2) We have included the more detailed description in the “Materials and Methods” section. “Gold NanoTech Inc. utilizes unique, and patented technology to physically break down bulk gold into nanoparticles, followed by epitaxially stacking these nanoparticles into stacked materials of controlled diameter within the nanometric range. Gold nanoparticle produced by this manufacturing procedure possesses distinctive physical properties owing to a special ionic charge that maintains its structure and is different from commercially available nanogold produced by chemical reduction methods.” (Page 3, line 141-147)
And we also reworded the Title: “Evaluation of the Biocompatibility and Endothelial Differentiation Capacity of Mesenchymal Stem Cells by Polyethylene Glycol Nanogold Composites”

2- Title; Polyethylene glycol not glycerol.
Answer: Thanks for the correction from the Reviewer. We have reworded the name of the title. “Evaluation of the Biocompatibility and Endothelial Differentiation Capacity of Mesenchymal Stem Cells by Polyethylene Glycol Nanogold Composites”

3- Abstract, line 22; MSCs have been incorporated into the nanoparticles? This is not the case here.
Answer: We have reworded the description. “The combination of Mesenchymal Stem Cells (MSCs) and nanoparticles are potential nanomaterials, …” (Page 1, line 22-23)

4- Page 7, line 318; What does decoration mean? Later UV-Vis and FTIR results contradict the hypothesis of surface decoration when the absorbance intensity declines. 
Answer:
(1) We have reworded the decoration to “presence”. “UV-Vis spectroscopy was applied to confirm the presence of Au nanoparticles on PEG.” (Page 7, line 341-342)
(2) We have repeated and included the new data of UV-Vis and FTIR in Figure 1A & 1B. The new description is also included in “Results” section. “The specific peaks of pure PEG were at 2902.59 cm-1 (-CH2 stretching), 1645.84 cm-1 (-CH2 scissoring) and 1,097.33 cm-1 (C-O-C stretching). These peaks were also found in the PEG-Au 43.5 ppm group. However, the specific peaks of Au nanoparticles were at 3430.72 cm-1. In PEG-Au 43.5 ppm group, the peak of CH2 stretching, CH2 scissoring and C-O-C stretching was shifted to 2902.75 cm-1, 1646.32 cm-1 and 1097.63, respectively. The above evidence indicated the modification of Au nanoparticles on PEG nanofilm.” (Page 7, line 345-351)

5-FTIR results; I was wondering if no peak shift or disappearance takes place after Au inclusion? If so, how these components interact (bond)?
Answer:
We have included the new FTIR results and description in Figure 1B, Results section, and Figure caption 1B.
“Next, the FTIR spectra demonstrated the functional groups in Au nanoparticles, pure PEG and PEG incorporated with 43.5 ppm of Au nanoparticles (Figure 1B). The specific peaks of pure PEG were at 2902.59 cm-1 (-CH2 stretching), 1645.84 cm-1 (-CH2 scissoring) and 1,097.33 cm-1 (C-O-C stretching). These peaks were also found in the PEG-Au 43.5 ppm group. However, the specific peaks of Au nanoparticles were at 3430.72 cm-1. In PEG-Au 43.5 ppm group, the peak of CH2 stretching, CH2 scissoring and C-O-C stretching was shifted to 2902.75 cm-1, 1646.32 cm-1 and 1097.63, respectively. The above evidence indicated the modification of Au nanoparticles on PEG nanofilm” (Page 7, line 345-351)
(B) The FTIR spectra of Au nanoparticles, pure PEG and PEG-Au 43.5 ppm.” (Page 8, line 381-382)

6- What is the ROS scavenging mechanism?
Answer:
DPPH (2,2-diphenyl-1-picrylhydrazyl) produces purple color in methanol solution, and it will fade to shade of yellow color in the presence of antioxidants. The reaction of decolorization occurred when odd electron of the nitrogen atom in DPPH is reduced by receiving hydrogen atom from antioxidant compounds [1].
Reference:
1. Scherer, R.; Godoy, H.T. Antioxidant activity index (AAI) by the 2, 2-diphenyl-1-picrylhydrazyl method. Food chemistry 2009, 112, 654-658.

We have also included the reference described for DPPH in section 2.3. “The free radical scavenging ability of the pure PEG and PEG-Au 43.5 ppm nanocomposites were measured using DPPH assay (2,2-diphenyl-1-picrylhydrazyl) (Sigma-Aldrich, Burlington, MA, USA) which was described in previous study [32].” (Page 4, line 171-173)

7- The FTIR spectra once again imply the presence of the Au NPs within the polymer, solely all PEG peaks are shown in both spectra. If so, why water contact angle declines despite entrapment of Au NPs within the polymer.
Answer:
The Au nanoparticle is no molecular vibration and movement. Therefore, the absorption peak cannot be observed on the FTIR spectrum. However, Au nanoparticle can generate intermolecular forces with hydroxyl groups, an absorption peak can be observed at 3430.7 cm-1 on FTIR spectrum. Based on above result, so it can increase the hydrophilicity of the film surface.

8- Figure 1; What is the control?
Answer:
(1) The deionized water represented as the control group of DPPH assay. We also included more description in section 2.3. “The absorbance value of test sample and the control represented as nanomaterials and deionized water, respectively.” (Page 4, line 179-180) And in Figure 1D caption: “Deionized water represented as the control group.” (Page 8, line 385)
(2) The pure Au was the control group for water contact angle measurement. More information was included in Figure 1E caption: “Pure Au nanoparticles were represented as the control group.” (Page 8, line 387-388)
and in section 2.4. “Pure Au nanoparticles were used as the control group.” (Page 4, line 186)

9- Were the samples sterilized? How?

Answer:

Thanks the valuable comment from reviewer. Sample preparation was under aseptic condition. Therefore, it does not need to be sterilized.

10-The optimum interaction of MSC reflected in their larger F actin filaments could be related to biomechanics of the substrate. PEG-Au is assumed to be stiffer than PEG and this stiffness discrepancy could lead to formation of larger F-actin filaments. How the chemical and mechanical effects could be separately analysed and discussed?
Answer:
In our study, there is no difference in the mechanical properties of the substrate. However, it can be known from the XPS analysis that there are exposed gold nanoparticles on the surface. Therefore, it can be considered that the chemical effect has a greater impact.

11- Page 9, last paragraph; How surface nanotopography and surface charge play a role in cell-material interaction?
Answer:
In previously study show that gold nanoparticles in small amounts induced significant changes in PU surface morphology and domain structures, from hard segment lamellae to soft segment micelles. These changes resembled the morphological transformation among different mesophases occurred in diblock copolymers.  Better cellular proliferation and lower platelet activation were demonstrated for the PU nanocomposite with 43.5 or 65 ppm of Au than the pure PU or the nanocomposite containing a different amount of Au [1]. In our study, the images exhibited that the pure PEG was homogenous and uniform. However, the surface morphology became strip-like while Au nanoparticles were decorated onto PEG (see Figure 1C). The above result show that gold nanoparticle in nanocomposite can create various nano-topography and induced cell adhesion and proliferation.
Reference:
1.     Hsu, S.-h.; Tang, C.-M.; Tseng, H.-J. Gold nanoparticles induce surface morphological transformation in polyurethane and affect the cellular response. Biomacromolecules 2008, 9, 241-248.

12- Figure 5A, merge with what?
Answer:
We have changed the color of “Merge” to black color, and also included the description on Figure 5A caption. “The images were merged with DAPI stained cell nuclei.” (Page 12, line 466)

13- Discussion, 1st paragraph; English editing is needed for this part of discussion.
Answer:
We have reworded the 1st paragraph in Discussion section. “Blood vessels are highly complicated in structure which lack of regeneration capacity. Various functions of the blood vessels are exhibited through the presence of the Extracellular Matrix (ECM). The different composition, thickness, and structure of ECM support the formation of blood vessels such as arteries, veins, and capillaries [37]. However, due to lack of ECM which can provide support for cell proliferation, migration and attachment, creating vascular conduits to mimic native blood vessels is difficult. ECM is also a major component in the process of angiogenesis [38]. Vascular replacement or revascularization are the standard procedures for clinical surgeries. However, autologous grafts are commonly sufficient due to limited vessel availability and difficult to be acquired. Thus, the demand for appropriate vascular grafts with long-term patency has gradually increased [39]. Surface modification through chemical, physical, and biomolecule immobilization have been well investigated to improve the biocompatibility of polymeric grafts. Previous literature has also verified that modifying the surface properties of biomaterials at the nano-scale level can enhance cell adhesion capacity [40]. For instance, the surface of PLGA film was formulated into spherical 200 nm features, which could facilitate endothelial cell adhesion as compared to smooth PLGA and 100 nm or 500 nm PLGA surface features [41]. In current study, the homogenous feature of pure PEG was observed to become strip-like after incorporating with 43.5 ppm Au nanoparticles. Furthermore, hydrophilic measurement also indicated PEG-Au 43.5 ppm could enhance cell adhesion capacity” (Page 14-15, line 514-532)

14- Page 15, line 511-523; This paragraph has nothing to do with the current study and does not justify the results related to PEG-Au for surface coating of vascular grafts.
Answer:
Thanks for the valuable comment from the Reviewer. We have removed the original paragraph and included a new description in Discussion section. “As mentioned above, the nano-scale surface modified with nanoparticles under an appropriate condition could provide focal points for filopodia of cells to attach more efficiency [42]. Therefore, the noble metals such as Au and silver (Ag) nanoparticles have attracted the attention due to their special characteristics. For instance, Ag nanoparticles have verified to be a potential anti-microbial nanomaterial [43]. Our research team have developed the nanocomposites fabricating Ag nanoparticles with Col, which demonstrated the longer stress fiber extension for MSCs adhesion and migration [44]. Additionally, Au nanoparticles were proved to be non-cytotoxic, which owned superior biocompatibility for biomedical applications such as vascular regeneration [45]. A piece of literature also described the Col-Au nanomaterial significantly stimulated MSCs to have more protrusions (filopodia) and elongation for attachment [25]. In line with current results, the PEG nanofilm modified with 43.5 ppm of Au nanoparticles was observed to promote the expression of vinculin and elongate F-actin fiber, supporting that PEG-Au 43.5 ppm could significantly influence cell morphology to promote MSCs attachment.” (Page 15, line 533-546)

15- line 525-545, What this literature review is to imply? Any relevance to the results to be discussed?
Answer:
We have reworded the description in Discussion section. “Promoting angiogenesis and inhibiting thrombosis are major factors in vascular regeneration. Recent research study created a vascular graft composed of polycaprolactone-methoxy polyethylene glycol as the inner layer and poly [4,4’-methylenebis (phenyl isocyanate)-alt-1,4-butanediol/di(propylene glycol)/polycaprolactone] (PU-PCL) as the outer layer. Focusing on the inner layer of the graft, polycaprolactone-methoxy polyethylene glycol significantly improved the hydrophilicity and anti-thrombogenic properties when compared to PCL or PU-PCL membranes. Additionally, based on in vivo assessment, the polycaprolactone-methoxy polyethylene glycol membrane strengthened the differentiation of rat bone marrow stem cells into ECs, and also significantly facilitated the protein and mRNA gene expression of CD31, Flk-1, and vWF, which was associated with angiogenesis [48]. Our previous study also demonstrated Col fabricated with 43.5 ppm of Au nanoparticles significantly enhanced CD31 expression as well as promoted endothelial differentiation for MSCs [25], which was corresponded to Figure 5 in current research.” (Page 15, line 557-569)

16- Line 546-547; what this sentence means? In general, discussion is indeed poor and vague.
Answer:
(1) We have reworded the sentence and moved the paragraph to appropriate place in Discussion section. “Literature indicated Au nanoparticles functionalized nanoparticles had efficient radical scavenging property [46], which was corresponded to the data for PEG-Au 43.5 ppm nanocomposites (Figure 1D).” (Page 15, line 547-549)
(2) Thanks for the comment from the Reviewer. We have reworded the description in Discussion section.

17-line 556, or surface topography? no other change took place, e.g. mechanical stiffness, surface charge, etc.?
Answer:
(1) We have reworded the description “Above all, our findings indicate that the modification of 43.5 ppm Au nanoparticles on PEG nanofilm lead to better surface topography, thus promoting cell attachment.” (Page 16, line 583-584)
(2) And we also included our previous studies and description in Discussion section for the verification of mechanical properties improved by different concentration of Au nanoparticles. “The biomechanical and haemodynamic properties of vascular grafts are highly concerned. The surface of Col [31] and FN [49] nanofilm modified with different concentration (17.4, 43.5, and 174 ppm) of Au nanoparticles were discussed in our previous studies. The Young’s modulus analysis demonstrated the elasticity of Col-Au and FN-Au nanocomposites were decreased with the increased concentration of Au. Furthermore, the surface roughness observed through AFM evaluated the thicker fibrils due to more collagen aggregation caused by 43.5 ppm of Au nanoparticles. The measurement of hydrophilicity property demonstrated the increased concentration of Au lead to better wettability for Col-Au and FN-Au nanocomposites. Additionally, the thermogravimetric analysis (TGA) figured out the pyrolytic temperatures were increased with the addition of 43.5 ppm Au nanoparticles into Col and FN. Based on the above evidences, we inference the PEG nano film modified with the optimal concentration of Au (43.5 ppm) will improve mechanical resistance to the blood flow.” (Page 15-16, line 570-582)

Reviewer 2 Report

Introduction

Please rewrite this paragraph because you repeated the ideas (in violet).

Lines 47 to 50 - "Blood vessels play the important role of being conduits for blood [1]. Furthermore, a body’s cardiovascular system is essential for life. The cardiovascular system is responsible for oxygen, carbon dioxide and nutrient transportation within the human body, while the vasculature in the cardiovascular system is the conduit for blood flow [2]."

Line 69, 72, 77, 504, 513 - So many definitions of Poly(Lactic‐co‐glycolic acid) (PLGA) as Polyethylene Glycol (PEG) lines 74, 78, 136 and 137; Endothelial Cells Lines 93 and 110; Mesenchymal Stem Cells (MSCs) in lines 97, 113 and in the abstract; von Willebrand Factor (vWF) lines 119 and 539;

Line 120 - (I’m not sure about my editing at the end of this sentence) ????

Materials and Methods

More information should be provided about the animal model, namely that related to the euthanasia of rats.

After being given local anesthesia, the dorsal skin of each rat was incised at 10 mm to subcutaneously implant the pure PEG and PEG‐Au 43.5 ppm nanocomposites. After 4 weeks, rats were euthanized (????) and the wound tissue was resected for further experiments.

Please provide more information and better describe this "The fibrous capsule formation in six sites was investigated through Hematoxylin and Eosin (H&E) staining, with the average encapsulated fibrotic tissues being quantified by commercial software."

Line 307 - Please define DAPI.

There must be consistency and correspondence between the sequence of analyzes in the Materials and Methods section and that of results.

Results

How did you assess the thickness of the capsule?

I could not agree with yours description of the histopathological results.

In Figure 6A, the formation of a fibrous capsule was detected and its thickness/area (????) was measured in the context of a histopathological evaluation of processed skin samples that were stained with H&E.

Additionally, we also investigated the endothelialization marker CD31 in tissue through IHC staining, with the images displayed in Figure 6B."

Lines 454-455 I could not agree with this, please explain better this results in terms of histopathology analysis.

In figure 6 please explain the differences between the 3 groups, I couldn't find them,...

Discussion

Line 492 - The ECM in arteries,  veins, and capillaries

References

25 - Hung, H.‐S.; Chang, C.‐H.; Chang, C.‐J.; Tang, C.‐M.; Kao, W.‐C.; Lin, S.‐Z.; Hsieh, H.‐H.; Chu, M.‐Y.; Sun, W.‐S.; Hsu, S.‐h. In vitro study of a novel nanogold‐collagen composite to enhance the mesenchymal stem cell behavior for vascular regeneration. PLoS One 2014, 9, e104019.

34 - Hung, H.‐S.; Chang, C.‐H.; Chang, C.‐J.; Tang, C.‐M.; Kao, W.‐C.; Lin, S.‐Z.; Hsieh, H.‐H.; Chu, M.‐Y.; Sun, W.‐S.; Hsu, S.‐h. J.P.O. In vitro study of a novel nanogold‐collagen composite to enhance the mesenchymal stem cell behavior for vascular regeneration. 2014, 9, e104019.

Different numbers but the same reference !!!!!!

Author Response

Comments and Suggestions for Authors

Introduction

Please rewrite this paragraph because you repeated the ideas (in violet).

Lines 47 to 50 - "Blood vessels play the important role of being conduits for blood [1]. Furthermore, a body’s cardiovascular system is essential for life. The cardiovascular system is responsible for oxygen, carbon dioxide and nutrient transportation within the human body, while the vasculature in the cardiovascular system is the conduit for blood flow [2]."
Answer:
Thanks for the valuable suggestion from the Reviewer. We have reworded this paragraph.
“Blood vessels such as arteries, veins, and capillaries, are necessary for blood flow in human body [1]. The cardiovascular system is responsible for oxygen, carbon dioxide and nutrient transportation within the human body [2]. Therefore, the body’s cardiovascular system is essential for life.” (Page 2, Line 47-50)

Line 69, 72, 77, 504, 513 - So many definitions of Poly(Lactic‐co‐glycolic acid) (PLGA) as Polyethylene Glycol (PEG) lines 74, 78, 136 and 137; Endothelial Cells Lines 93 and 110; Mesenchymal Stem Cells (MSCs) in lines 97, 113 and in the abstract; von Willebrand Factor (vWF) lines 119 and 539;
Answer:
Thanks for the comment from the Reviewer. We have removed some definitions in the manuscript to make it more concise.

Line 120 - (I’m not sure about my editing at the end of this sentence) ????
Answer:
We have removed the sentence.

Materials and Methods

More information should be provided about the animal model, namely that related to the euthanasia of rats.
Answer:
We have included the information for our animal experiments. “Female Sprague Dawley (SD) rats (age: 2-3 months, weight: 300 – 350 g) were obtained from National Laboratory Animal Center (Taipei, Taiwan)” (Page 7, Line 312-313)
“The Olympus ix71 fluorescence microscope (Tokyo, Japan) was applied to further analysis the fluorescence intensity.” (Page 7, Line 327-328)

After being given local anesthesia, the dorsal skin of each rat was incised at 10 mm to subcutaneously implant the pure PEG and PEG‐Au 43.5 ppm nanocomposites. After 4 weeks, rats were euthanized (????) and the wound tissue was resected for further experiments.
Answer:
We have reworded the description. “In vivo experiments, PEG and PEG-Au composite solution was coated on glass coverslip (15 mm) and implanted to rat subcutaneous tissue. After being given local anesthesia, the nanocomposites were subcutaneously implanted into 10 mm2 incised dorsal skin area of rats for 4 weeks. The rats (n=5) were then sacrificed, and the tissues with implanted materials were further subjected to histological investigations.” (Page 7, Line 314-319)

Please provide more information and better describe this "The fibrous capsule formation in six sites was investigated through Hematoxylin and Eosin (H&E) staining, with the average encapsulated fibrotic tissues being quantified by commercial software."
Answer:
Thanks for the valuable comment form the Reviewer. We have included more information. “The formation of fibrous capsule in six sites was investigated by Hematoxylin and Eosin (H&E) staining, and the thickness of encapsulated fibrotic tissues was calculated and quantified by Image J 5.0 software.” (Page 7, Line 319-321)

Line 307 - Please define DAPI.
Answer:
We have included the definition about DAPI. “DAPI (4',6-diamidino-2-phenylindole) staining was used to target the nuclear DNA in fixed cells.” (Page 7, Line 328-329)

There must be consistency and correspondence between the sequence of analyzes in the Materials and Methods section and that of results.
Answer:
Thanks for the valuable suggestion from the Reviewer. We have included new description in Materials and Methods section. “2.7. Cytoskeleton of MSCs. Cells at the density of 1 × 104 per well were seeded in a 24-well plate coated with various materials for 24 hours incubation. First, MSCs were fixed with 4% paraformaldehyde (PFA) for 10 min, permeated with 0.5% Triton X-100 in PBS then reacted for 10 min at room temperature (RT). Next, phalloidin (~6 mM) (Sigma, USA) was treated for 60 min in dark at RT. Ultimately, 4, 6-diamidion-2-phenylindole (DAPI) (Invitrogen, White Plains, NY, USA) solution was used to stain the cell nucleus in dark for 10 mins. The cytoskeletal of MSCs were observed by a fluorescent microscope [25].” (Page 5, line 252-259)

Results

How did you assess the thickness of the capsule?
Answer:
We used Image J 5.0 software to calculate the average thickness of fibrous capsule in the implanted tissues. We have also used white arrows to demonstrate the thickness of capsule in Figure 6A (Page 13, line 489). The more description was included in Material and Methods section. “The formation of fibrous capsule in six sites was investigated by Hematoxylin and Eosin (H&E) staining, and the thickness of encapsulated fibrotic tissues was calculated and quantified by Image J 5.0 software.” (Page 6, Line 319-321)

I could not agree with yours description of the histopathological results.
Answer:
We have reworded the description of in vivo assessments for histopathological results.
“The nanomaterials, pure PEG and PEG-Au 43.5 ppm, were subcutaneously implanted into our rat models for 4 weeks to evaluate the in vivo biocompatibility. The anti-inflammatory effect and endothelial differentiation capacity were further investigated after 4 weeks of implantation. In Figure 6A, the effect of each nanomaterial on fibrotic encapsulation in tissues were analyzed through H&E staining.” (Page 12-13, Line 467-471)
“The above results elucidated PEG-Au 43.5 ppm significantly attenuated foreign body responses as well as induced endothelialization in vivo assessments for tissue regeneration.” (Page 13, Line 479-480)
“The above result indicated both pure PEG and PEG-Au 43.5ppm would not harm the surrounding tissues after implantation.” (Page 14, Line 499-500)

In Figure 6A, the formation of a fibrous capsule was detected and its thickness/area (????) was measured in the context of a histopathological evaluation of processed skin samples that were stained with H&E.
Answer:
We have reworded the description. “In Figure 6A, the effect of each nanomaterial on fibrotic encapsulation in tissues are analyzed through H&E staining.” (Page 12-13, Line 476-477)

Additionally, we also investigated the endothelialization marker CD31 in tissue through IHC staining, with the images displayed in Figure 6B."
Answer:
We have modified the description. “Additionally, the expression of endothelial marker CD31 in tissue was also investigated through IHC staining, with the images displayed in Figure 6B.” (Page 13, line 480-482)

Lines 454-455 I could not agree with this, please explain better this results in terms of histopathology analysis.
Answer:
We have reworded the description in Results 3.5.. “The nanomaterials, pure PEG and PEG-Au 43.5 ppm, were subcutaneously implanted into our rat models for 4 weeks to evaluate the in vivo biocompatibility. The anti-inflammatory effect and endothelial differentiation capacity were further investigated after 4 weeks of implantation.” (Page 12, Line 473-476)

In figure 6 please explain the differences between the 3 groups, I couldn't find them,...
Answer:
We have included the explanation for the effects of nanomaterials.
“The above results elucidated PEG-Au 43.5 ppm significantly attenuated foreign body responses as well as induced endothelialization in vivo assessments for tissue regeneration.” (Page 13, Line 484-486)
And also for Figure 7 in Results section. “The above result indicated both pure PEG and PEG-Au 43.5ppm would not harm the surrounding tissues after implantation.” (Page 14, Line 499-500)

Discussion

Line 492 - The ECM in arteries,  veins, and capillaries
Answer:
We have reworded the sentence. “The different composition, thickness, and structure of ECM support the formation of blood vessels such as arteries, veins, and capillaries [37].” (Page 14, Line 516-517)

References

25 - Hung, H.‐S.; Chang, C.‐H.; Chang, C.‐J.; Tang, C.‐M.; Kao, W.‐C.; Lin, S.‐Z.; Hsieh, H.‐H.; Chu, M.‐Y.; Sun, W.‐S.; Hsu, S.‐h. In vitro study of a novel nanogold‐collagen composite to enhance the mesenchymal stem cell behavior for vascular regeneration. PLoS One 2014, 9, e104019.

34 - Hung, H.‐S.; Chang, C.‐H.; Chang, C.‐J.; Tang, C.‐M.; Kao, W.‐C.; Lin, S.‐Z.; Hsieh, H.‐H.; Chu, M.‐Y.; Sun, W.‐S.; Hsu, S.‐h. J.P.O. In vitro study of a novel nanogold‐collagen composite to enhance the mesenchymal stem cell behavior for vascular regeneration. 2014, 9, e104019.

Different numbers but the same reference !!!!!!

Answer:

Thanks for the correction from the Reviewer. We have removed the same reference in References section.

Reviewer 3 Report

The authors prepared scaffolds based on polyethylene glycol (PEG) and PEG incorporated with 43.5 ppm of gold nanoparticles (PEG-Au) in this manuscript. Both scaffolds were characterized physical-chemically, and their biological effects were assessed through many complementary in vitro and in vivo tests.

The study fits with the scope of Polymers and should be of interest to the journal’s readers. The methodology used by the authors is robust, and the experiments were carefully done. The manuscript is well written, and the presentation of results is scientifically sound and of quality. I have the following observation.

Major points:

  • It is not clear what these scaffolds are intended for. The introduction section focuses on vascular grafts and the importance of using materials to improve endothelization and prevent thrombosis; therefore, I suppose that is the intended use of the scaffolds developed in the present article. However, authors should better clarify and specify this aspect.
  • Following the point above, if the intended use of the developed PEG and PEG-Au is as a vascular graft, concerns about their mechanical resistance to the blood pressure are questioned. Therefore, the authors should also clarify this aspect.
  • It is not clear the exact nature of what authors prepared in their manuscript. I have classified PEG and PEG-Au as scaffolds, but I am not sure this is correct. From the preparation procedures described in section 2.1.2, they seem films. Authors should better clarify and specify this aspect. Maybe a photograph may help.
  • It is not clear what the control is in the following experiments: ROS-scavenging ability, contact angle, cell viability, ROS generation, platelet activation and monocyte conversion, evaluation of F-actin/vinculin expression and cell migration ability, endothelial differentiation, in vivo inflammation and endothelization. Please report what the control is both in the materials and methods section and in the caption of each figure.

Minor points:

  • Abstract, line 22: “nanoparticles containing mesenchymal stem cells…” this sentence seems inappropriate. Cells are too big to fix into nanoparticles, especially metallic ones.
  • Line 120: a comment from one author is present in the main text.
  • Line 324: it is not clear. If the specific peaks of PEG were observed in PEG-Au, they should confirm the presence of PEG and not of Au nanoparticles.

Author Response

Comments and Suggestions for Authors

The authors prepared scaffolds based on polyethylene glycol (PEG) and PEG incorporated with 43.5 ppm of gold nanoparticles (PEG-Au) in this manuscript. Both scaffolds were characterized physical-chemically, and their biological effects were assessed through many complementary in vitro and in vivo tests.

The study fits with the scope of Polymers and should be of interest to the journal’s readers. The methodology used by the authors is robust, and the experiments were carefully done. The manuscript is well written, and the presentation of results is scientifically sound and of quality. I have the following observation.

Major points:

  • It is not clear what these scaffolds are intended for. The introduction section focuses on vascular grafts and the importance of using materials to improve endothelization and prevent thrombosis; therefore, I suppose that is the intended use of the scaffolds developed in the present article. However, authors should better clarify and specify this aspect.

Answer:

Thanks for the valuable suggestion from the Reviewer. We have included the new description in Discussion section.

(1) “As mentioned above, the nano-scale surface modified with nanoparticles under an appropriate condition could provide focal points for filopodia of cells to attach more efficiency [42]. Therefore, the noble metals such as Au and silver (Ag) nanoparticles have attracted the attention due to their special characteristics. For instance, Ag nanoparticles have verified to be a potential anti-microbial nanomaterial [43]. Our research team have developed the nanocomposites fabricating Ag nanoparticles with Col, which demonstrated the longer stress fiber extension for MSCs adhesion and migration [44]. Additionally, Au nanoparticles were proved to be non-cytotoxic, which owned superior biocompatibility for biomedical applications such as vascular regeneration [45]. A piece of literature also described the Col-Au nanomaterial significantly stimulated MSCs to have more protrusions (filopodia) and elongation for attachment [25]. In line with current results, the PEG nanofilm modified with 43.5 ppm of Au nanoparticles was observed to promote the expression of vinculin and elongate F-actin fiber, supporting that PEG-Au 43.5 ppm could significantly influence cell morphology to promote MSCs attachment.” (Page 15, Line 533-546)

(2) “Literature indicated Au nanoparticles functionalized nanoparticles had efficient radical scavenging property [46], which was corresponded to the data for PEG-Au 43.5 ppm nanocomposites (Figure 1D). Furthermore, in line with our results, after 43.5 ppm of Au nanoparticles were decorated on the synthetic polymer PEG, the activation of platelets and monocytes, as well as the ROS production, significantly decreased (Figure 3). The interactions between monocytes and platelets could enhance monocyte migration into blood vessel walls causing atherogenesis [47]. Thus, the appropriate addition of Au nanoparticles can remarkably facilitate cell viability and anti-oxidative capacity for MSCs. As mentioned above, MSCs could provide various functions such as secreting cytokines for tissue repair. MSCs can also be induced to differentiate into EC-like cells [27].” (Page 15, Line 547-556)

(3) “Promoting angiogenesis and inhibiting thrombosis are major factors in vascular regeneration. Recent research study created a vascular graft composed of polycaprolactone-methoxy polyethylene glycol as the inner layer and poly [4,4’-methylenebis (phenyl isocyanate)-alt-1,4-butanediol/di(propylene glycol)/polycaprolactone] (PU-PCL) as the outer layer. Focusing on the inner layer of the graft, polycaprolactone-methoxy polyethylene glycol significantly improved the hydrophilicity and anti-thrombogenic properties when compared to PCL or PU-PCL membranes. Additionally, based on in vivo assessment, the polycaprolactone-methoxy polyethylene glycol membrane strengthened the differentiation of rat bone marrow stem cells into ECs, and also significantly facilitated the protein and mRNA gene expression of CD31, Flk-1, and vWF, which was associated with angiogenesis [48]. Our previous study also demonstrated Col fabricated with 43.5 ppm of Au nanoparticles significantly enhanced CD31 expression as well as promoted endothelial differentiation for MSCs [25], which was corresponded to Figure 5 in current research.” (Page 15, Line 557-569)

(4) “The biomechanical and haemodynamic properties of vascular grafts are highly concerned. The surface of Col [31] and FN [49] nanofilm modified with different concentration (17.4, 43.5, and 174 ppm) of Au nanoparticles were discussed in our previous studies. The Young’s modulus analysis demonstrated the elasticity of Col-Au and FN-Au nanocomposites were decreased with the increased concentration of Au. Furthermore, the surface roughness observed through AFM evaluated the thicker fibrils due to more collagen aggregation caused by 43.5 ppm of Au nanoparticles. The measurement of hydrophilicity property demonstrated the increased concentration of Au lead to better wettability for Col-Au and FN-Au nanocomposites. Additionally, the thermogravimetric analysis (TGA) figured out the pyrolytic temperatures were increased with the addition of 43.5 ppm Au nanoparticles into Col and FN. Based on the above evidences, we inference the PEG nano film modified with the optimal concentration of Au (43.5 ppm) will improve mechanical resistance to the blood flow.” (Page 15-16, Line 570-582)

  • Following the point above, if the intended use of the developed PEG and PEG-Au is as a vascular graft, concerns about their mechanical resistance to the blood pressure are questioned. Therefore, the authors should also clarify this aspect.
    Answer:
    We have included the new description for biomechanical properties in “Discussion” section.
    “The biomechanical and haemodynamic properties of vascular grafts are highly concerned. The surface of Col [31] and FN [49] nanofilm modified with different concentration (17.4, 43.5, and 174 ppm) of Au nanoparticles were discussed in our previous studies. The Young’s modulus analysis demonstrated the elasticity of Col-Au and FN-Au nanocomposites were decreased with the increased concentration of Au. Furthermore, the surface roughness observed through AFM evaluated the thicker fibrils due to more collagen aggregation caused by 43.5 ppm of Au nanoparticles. The measurement of hydrophilicity property demonstrated the increased concentration of Au lead to better wettability for Col-Au and FN-Au nanocomposites. Additionally, the thermogravimetric analysis (TGA) figured out the pyrolytic temperatures were increased with the addition of 43.5 ppm Au nanoparticles into Col and FN. Based on the above evidences, we inference the PEG nano film modified with the optimal concentration of Au (43.5 ppm) will improve mechanical resistance to the blood flow.” (Page 15-16, Line 570-582)

  • It is not clear the exact nature of what authors prepared in their manuscript. I have classified PEG and PEG-Au as scaffolds, but I am not sure this is correct. From the preparation procedures described in section 2.1.2, they seem films. Authors should better clarify and specify this aspect. Maybe a photograph may help.
    Answer:
    In our study, PEG and PEG-Au solutions were coated on the surface of culture dishes to form a thin film. The samples analyzed in the following Biocompatibility assay are all prepared through the above methods.
    We have also included the more detail description in the “Materials and Methods” section 2.6. Biocompatibility assay, “PEG and PEG-Au solution was coated onto culture dish to perform the following evaluation” (Page 5, line 206-207); in the section 2.11. Rat Subcutaneous Implantation, “In vivo experiments, PEG and PEG-Au composite solution was coated on glass coverslip (15 mm) and implanted to rat subcutaneous tissue.” (Page 6, line 314-316)

  • It is not clear what the control is in the following experiments: ROS-scavenging ability, contact angle, cell viability, ROS generation, platelet activation and monocyte conversion, evaluation of F-actin/vinculin expression and cell migration ability, endothelial differentiation, in vivo inflammation and endothelization. Please report what the control is both in the materials and methods section and in the caption of each figure.

Answer:
We have included the description for the control group in each experiment, and we also modified each figure to make it to be more clear.

In “Materials & Methods” section:
2.3. “The absorbance value of test sample and the control represented as nanomaterials and deionized water, respectively.” (Page 4, line 179-180)
2.4. “Pure Au nanoparticles were used as the control group.” (Page 4, line 186)
2.6.1. “The MSC alone group represented as the control.” (Page 5, line 208)
2.6.2. “The MSC alone group expressed as the control group.” (Page 5, line 219-220)
2.6.3. “The monocyte alone exhibited as the control group.” (Page 5, line 232-233) and “The platelet alone group exhibited as the control.” (Page 5, line 238)

2.8. “The MSC alone represented as the control.” (Page 6, line 270)

2.9. “The MSC alone exhibited as the control group.” (Page 6, line 275-276)

2.10. “The MSC alone presented as the control group.” (Page 6, line 290)

2.11. “The MSC alone represented as the control.” (Page 7, line 329-330)

In “Figure caption”:

Figure 1 “(D) …Deionized water represented as the control group….” (Page 8, line 385) And “(E) Pure Au nanoparticles represented as the control group.” (Page 8, line 387-388)

Figure 2 “*p < 0.05, **p < 0.01: compared to the MSC alone group. #p < 0.05: compared to the pure PEG group.” (Page 9, line 395 and 399)

Figure 3 “*p < 0.05, **p < 0.01: compared to the Platelet/Monocyte alone group. #p < 0.05: compared to the pure PEG group.” (Page 10, line 428)

Figure 4 “*p < 0.05, **p < 0.01: compared to the MSC alone group. #p < 0.05: compared to the pure PEG group.” (Page 11, line 451-452)
Figure 5 “**p < 0.01: compared to the MSC alone group. #p < 0.05: compared to the pure PEG group.” (Page 12, line 470-471)

Figure 6 “*p < 0.05, **p < 0.01: compared to the MSC alone group. #p < 0.05: compared to the pure PEG group.” (Page 14, line 494)

 Minor points:

  • Abstract, line 22: “nanoparticles containing mesenchymal stem cells…” this sentence seems inappropriate. Cells are too big to fix into nanoparticles, especially metallic ones.
    Answer:

Thanks for the valuable suggestion from the Reviewer. We have reworded the description in Abstract. “The combination of Mesenchymal Stem Cells (MSCs) and nanoparticles are potential nanomaterials,…” (Page 1, line 22-23)

  • Line 120: a comment from one author is present in the main text.

Answer:

Thanks for the comment from the Reviewer. We have removed the sentence.

  • Line 324: it is not clear. If the specific peaks of PEG were observed in PEG-Au, they should confirm the presence of PEG and not of Au nanoparticles.
    Answer:

The Au nanoparticle is no molecular vibration and movement. Therefore, the absorption peak cannot be observed on the FTIR spectrum. However, Au nanoparticle can generate intermolecular forces with hydroxyl groups, an absorption peak can be observed at 3430.72 cm-1 on FTIR spectrum. However, the weakly adsorption peak easily covered with adsorption peak of OH group in PEG. In line 323, The sentence “These peaks were also found in the PEG-Au 43.5 ppm group, indicating the presence of Au nanoparticles.” was modified to “These peaks were also found in the PEG-Au 43.5 ppm group. However, the specific peaks of Au nanoparticles were at 3430.72 cm-1

Round 2

Reviewer 1 Report

Dear Editor,

Considering the applied corrections, I recommend the paper for publication.

Author Response

We appreciate your kindness comment.

Reviewer 3 Report

I am satisfied with the revision provided. However, the following sentence in the abstract “The combination of Mesenchymal Stem Cells (MSCs) and nanoparticles are potential nanomaterials, and their use is a promising strategy for treatment surrounding vascular regeneration” has no significance, as cells cannot be considered as nanomaterials.

Author Response

I am satisfied with the revision provided. However, the following sentence in the abstract “The combination of Mesenchymal Stem Cells (MSCs) and nanoparticles are potential nanomaterials, and their use is a promising strategy for treatment surrounding vascular regeneration” has no significance, as cells cannot be considered as nanomaterials.

Answer:

We appreciate your comment and valuable suggestion. We have reworded the description in Abstract. (Page 1, line 22-24)

“Mesenchymal Stem Cells (MSCs)-based treatment coupled with nanoparticles is considered to be a potential and promising therapeutic strategy for vascular regeneration.”